# Generalizable Diabetic Retinopathy Grading via Knowledge Constrained Concept Learning

## Abstract

Diabetic retinopathy (DR) grading models often suffer a significant performance drop when deployed to unseen clinical domains. A promising strategy is to mirror the diagnostic process of clinicians, who rely on identifying specific pathological signs to make judgments. Concept-based models (CBMs) are well-suited for this, but their effectiveness often hinges on concept supervision, which is rarely available in medical imaging. To address this, we propose Knowledge Constrained Concept Learning (KCCL), a novel framework that achieves robust domain generalization through concept learning under knowledge constraints. We first curate DRL6k, a dataset of 6,000 fundus images with lesion annotations, and train a lesion detection model to provide concept supervision via knowledge distillation. However, directly using this supervision may introduce noise and inconsistencies. Therefore, KCCL employs a knowledge constraint mechanism: it leverages medical priors to correct implausible concept predictions and reduce the influence of those deviating from clinical expectations during distillation, while also directly penalizing the model for producing clinically inconsistent concept predictions. Extensive experiments on multiple unseen target datasets demonstrate that KCCL significantly outperforms state-of-the-art domain generalization and DR grading methods, achieving generalization by producing clinically coherent and interpretable predictions.

## 1 Introduction

Diabetic retinopathy (DR) is a leading cause of blindness among working-age adults worldwide, affecting over 100 million people globally (Dai et al., 2024; Cheung et al., 2010). Early detection and timely intervention are crucial for preventing irreversible vision loss, making automated screening systems a critical tool in managing this condition. Recent advancements in deep learning (DL) have achieved remarkable success in automated DR grading from fundus images, demonstrating performance comparable to human experts and offering the potential to significantly expand screening accessibility (Dai et al., 2021; 2024; He et al., 2021). However, a critical limitation emerged when these models are deployed across different clinical settings. Models trained on one dataset often exhibit significant performance drops when applied to images from other sources, due to variations in imaging protocols, patient demographics, and equipment specifications. This domain shift problem poses a major barrier to the widespread clinical adoption of DL-based DR screening systems (Che et al., 2023). To address this challenge, numerous domain generalization approaches have been developed (Xia et al., 2024; Bi et al., 2024; Atwany et al., 2022). These methods approach the problem from various angles, attempting to learn diagnostically relevant features that are invariant across different domains, and have shown promising results. However, they all operate under a common paradigm: learning invariances implicitly from data patterns rather than explicitly incorporating the clinical knowledge that guides human diagnosis. This observation leads us to consider an alternative approach: what if we could directly embed the diagnostic reasoning process that opthalmologists use into the model? Human experts diagnose DR by systematically identifying and interpreting pathological lesions such as microaneurysms and exudates (Li et al., 2019). This concept-driven reasoning process is inherently robust to domain shifts, as the pathological manifestations of retinal disease remain consistent across different imaging conditions.

But how can we effectively translate this reasoning process into a deep learning framework? Recently, concept-based models (CBMs) (Koh et al., 2020; Espinosa Zarlenga et al., 2022) have emerged as a promising paradim to address this challenge. By channeling information through an explicit concept layer, they offer a more transparent decision-making process, allowing us to customize the model's reasoning process to align with clinical expertise—for instance, using retinal lesions as intermediate concepts to mirror how ophthalmologists diagnose diabetic retinopathy based on pathological findings. Moreover, recent studies have demonstrated that CBMs can enhance generalization across different domains (Choi et al., 2024; Chowdhury et al., 2024). However, these models confront a fundamental challenge in the medical domain: the scarcity of samples that are simultaneously annotated with both concept-level and downstream task labels, which hinders their applicability in practice.

To mitigate the scarcity of concept annotations, label-free methods (Oikarinen et al., 2022) have been explored, yet they typically rely on general-purpose vision-language models (Radford et al., 2021) that lack sufficient domain-specific alignment for medical imaging. A more practical alternative is to employ knowledge distillation by training a teacher model on a lesion annotated dataset to generate pseudo-labels for the broader dataset. However, this approach carries a substantial risk because a teacher model often suffers from limited generalization capabilities. Consequently, blindly using its predictions as supervision signals propagates noise and errors to the CBM's concept layer. Crucially, we recognize that while explicit concept labels are absent, disease severity labels are available and are intrinsically linked to lesion concepts. This insight prompts us to move beyond naive distillation and leverage medical priors derived from disease-lesion correlations to guide a robust and logically consistent concept learning process.

Building on this, we present Knowledge Constrained Concept Learning (**KCCL**), a novel framework that enables concept learning through knowledge distillation and knowledge constraints. We first curate a consolidated dataset of 6,000 fundus images (DRL6k) with image-level annotations for four key DR-related lesion concepts from existing public datasets. We then train a lesion detection model on this curated dataset to provide concept supervision via knowledge distillation. To ensure the CBM learns robust and clinically reliable lesion concepts, we introduce a knowledge constraint mechanism. First, it refines the distillation guidance by validating the predictions of lesion detection model against established medical knowledge, simultaneously correcting implausible concept predictions and reweighting the distillation loss. This provides a clear "what-to-learn" signal for the concept layer. Second, it directly penalizes the CBM for predicting concept predictions that deviate from medical knowledge, providing a "what-is-wrong" signal that prevents the model from converging to clinically nonsensical solutions. By combining these mechanisms, our approach enables robust concept learning from limited lesion annotations, leading to a more generalizable and interpretable DR grading model. Our main contributions are summarized as follows:

- We propose KCCL, a novel domain generalization framework for DR grading that integrates concept-based reasoning with knowledge distillation, using a lesion detection model trained on our curated DRL6k dataset.

- We design a dual-constraint mechanism that incorporates medical constraints to simultaneously refine the teacher guidance and regularize the CBM concept predictions, ensuring clinically plausible concept learning.

- We conduct extensive experiments across multiple datasets, demonstrating that KCCL significantly outperforms state-of-the-art methods on unseen domains while benefiting from the interpretability of concept-based models.

## 2 RELATED WORK

**Domain Generalization.** Domain generalization aims to develop models that perform robustly across unseen domains, overcoming domain shift challenges. A significant body of work focuses on learning domain-invariant representations by minimizing the divergence between source domains (Muandet et al., 2013; Li et al., 2018b;a; Matsuura & Harada, 2020). Another bunch of methods approaches the problem from the data perspective, aiming to enrich the diversity of source domains through data augmentation or generative models to better cover potential target distributions (Zhou et al., 2020a;b; Mancini et al., 2020). Given the critical importance of generalization in clinical applications, there has been a growing interest in applying DG techniques to various medical

imaging tasks (Che et al., 2023; Atwany et al., 2022; Zhang et al., 2020; Wu et al., 2023b; Bi et al., 2024). However, these methods predominantly view medical images from purely a data-centric perspective, often neglecting the invaluable clinical expertise and medical knowledge. In contrast, our work addresses this gap by incorporating explicit medical knowledge into the learning process via concept-based reasoning, thereby enabling models that are not only robust to domain shifts but also more interpretable and align with clinical diagnostic processes.

**Concept-based Models.** Concept based models (CBMs) introduce an intermediate concept layer that forces models to make predictions through human-interpretable concepts, thereby enhancing model interpretability (Koh et al., 2020; Yuksekgonul et al., 2022; Espinosa Zarlenga et al., 2022; Zhang et al., 2024). A particularly relevant line of work focuses on label-free CBMs that aim to reduce the annotation burden by automatically discovering concepts without explicit supervision (Oikarinen et al., 2022; Moayeri et al., 2023; Tan et al., 2025; Gao et al., 2024), but these methods often rely on large language models or CLIP models (Radford et al., 2021), whose effectiveness is limited in the medical domain as they are not pre-trained with the specialized concepts. Notably, Pang et al. (2024) also proposed a method for integrating medical knowledge through CBMs, but this approach still requires additional annotation work from physicians, making it less applicable in practice. In contrast, our method alleviates the dependency on labels through knowledge distillation, and extracts medical knowledge priors from the concept labels themselves based on statistical methods, thus avoiding the costly expert annotation work.

**Knowledge Distillation.** Knowledge distillation (Hinton et al., 2015) is primarily used for model compression and acceleration, transferring knowledge from large models to smaller ones to enhance performance while reducing computational costs (Gou et al., 2021). Recent works have also explored concept distillation, but with fundamentally different motivations and methodologies from ours. For instance, Sousa et al. (2022) proposed a method to distill knowledge from a black-box model to train a concept-based proxy model, aiming to interpret the black-box model's decisions. Gupta et al. (2023) sought to leverage knowledge from pre-trained large models to assist in concept extraction, where concepts are represented as Concept Activation Vectors (Kim et al., 2018) rather than the outputs of a concept layer as in traditional CBMs. In contrast, our approach addresses the scarcity of concept annotations in medical scenarios, and we do not treat the teacher model as a completely reliable source of knowledge, allowing for a more flexible and robust integration of medical knowledge into the learning process.

**Medical Knowledge Integration.** Incorporating medical knowledge to guide model learning is a well-established strategy in medical imaging (Xie et al., 2021). Given the high cost and specialized expertise required for annotating medical images, leveraging prior knowledge is crucial for enhancing model learning efficiency and reliability. These approaches typically embed clinical priors into the model's architecture (Sun et al., 2021; Yu et al., 2021; Pang et al., 2024) or training objectives (Wu et al., 2023a; Zhang et al., 2023; Xie et al., 2019; Li et al., 2022). Despite their diversity, these methods often integrate knowledge in a static manner, directly constraining the model's representations or predictions. Our work extends this paradigm by employing clinical priors as a dynamic mediator within a knowledge distillation framework. Specifically, we leverage medical knowledge to correct obviously erroneous distillation targets, reweight potentially problematic supervision signals, and regularize the student's concept learning, enabling robust knowledge transfer.

## 3 METHOD

### 3.1 PROBLEM FORMULATION AND OVERVIEW

Given source domain $\mathcal{D}_s$ and target domains $\mathcal{D}_t$, where each sample consists of fundus image $x \in \mathcal{X}$ and DR severity grade $y \in \mathcal{Y} = \{0, 1, 2, 3, 4\}$ representing normal to proliferative DR. Our goal is to learn a robust model from $\mathcal{D}_s$ that can generalize well to $\mathcal{D}_t$ without requiring any target domain data. Following established benchmarks (Che et al., 2023), we train the model on a large-scale source domain and evaluate it on multiple target domains. While conventional deep learning models directly learn the mapping $x \to y$, CBMs introduce an interpretable $x \to c \to y$. Here, $c$ represents a set of intermediate, human-understandable clinical concepts (*e.g.*, microaneurysms, hemorrhages) that are first predicted from the image and then used to determine the final grade $y$. This approach has shown promise for domain generalization (Choi et al., 2024; Chowdhury et al., 2024). However, the training of concept layers requires paired concept annotations, which are scarce in medical domains.

To address this, we propose a Knowledge Constrained Concept Learning (KCCL) framework that integrates medical knowledge through three specialized mechanisms: **Self Correction** (SC) directly fixes concept predictions that clearly violate medical knowledge, **Distillation Reweighting** (DR) reduces the influence of potentially problematic samples that deviate from clinical patterns, and **Knowledge Constrained Regularization** (KCR) penalizes the concept layer for generating medically implausible concepts. The overall framework is illustrated in Figure 1. Below, we detail each component of the KCCL framework.

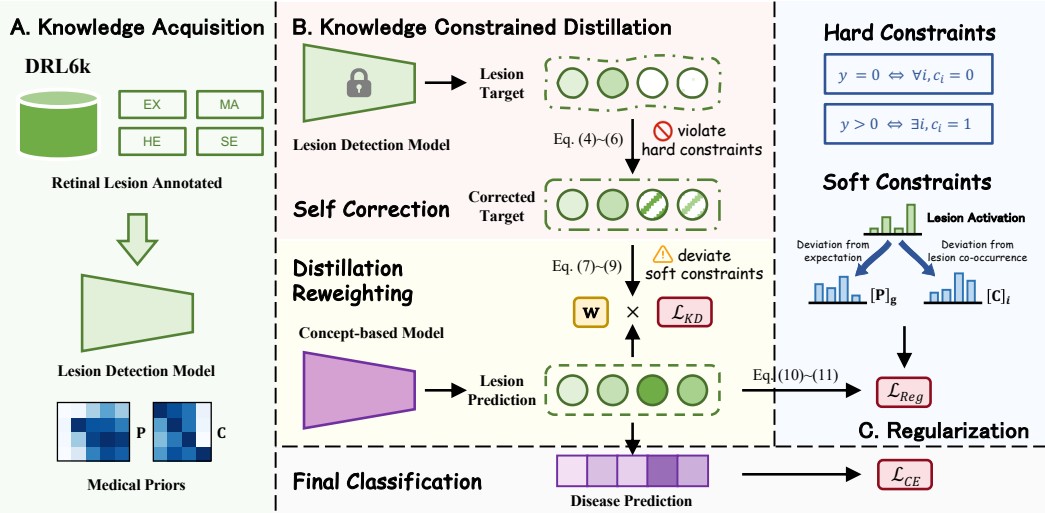

Figure 1: Overview of our KCCL framework. **A.** We first construct the DRL6k dataset to train a lesion detection model $T$ and derive medical priors, which inform the design of hard and soft constraints. **B.** Predictions from $T$ violating hard constraints are corrected via prior sampling; additionally, the distillation loss is dynamically reweighted based on the degree of soft constraint deviation. The final disease classification is based on these learned concepts. **C.** The CBM's concept layer is directly regularized to penalize violations of knowledge constraints.

Table 1: Statistics of the DRL6k Dataset.

| Dataset Split | Hard Exudates | Hemorrhages | Microaneurysms | Soft Exudates |
|---|---|---|---|---|
| Training | 2108 | 2733 | 2794 | 1084 |
| Validation | 265 | 347 | 358 | 144 |
| Test | 279 | 358 | 357 | 148 |

## 3.2 MEDICAL KNOWLEDGE ACQUISITION FOR GUIDANCE

**DRL6k.** We first construct the DRL6k dataset containing 6,000 images with image-level annotations for four representative and highly relevant DR-related lesions: Hard Exudates (EX), Soft Exudates (SE), Microaneurysms (MA), and Hemorrhages (HE). This dataset integrates FGADDR from (Wen et al., 2025), segmentation data from IDRID (Porwal et al., 2018), and the Retinal-Lesions dataset (Wei et al., 2021). We partition DRL6k into training, validation, and test sets using an 8:1:1 ratio, with detailed distribution statistics provided in Table 1.

**Lesion Detection Model Training.** Using this DRL6k, we train a lesion detection model $T$ using a standard ResNet50 (He et al., 2016) architecture on the DRL6k dataset. We train $T$ to predict concept probabilities $t$ for each image by minimizing the binary cross-entropy loss:

$$\mathcal{L}_T = -\frac{1}{K} \sum_{k=1}^{K} \left[ c_k \log(t_k) + (1 - c_k) \log(1 - t_k) \right], \tag{1}$$

where $t_k$ and $c_k$ are the predicted probability and binary label for concept $k$, respectively. Since $T$ serves only as an auxiliary knowledge source for subsequent concept learning, we adopt a straight-forward training approach without elaborate modifications (See Appendix for details).

**Knowledge Constraints Design.** To mitigate potential spurious correlations from $T$ and enforce clinically consistent concept learning, we incorporate two forms of prior knowledge as explicit constraints derived from established medical knowledge and statistical evidence. 1) *Hard Constraints* formalize basic medical rules: healthy retinas are lesion-free, while diseased retinas necessarily exhibit lesion presence. Violations of these constraints indicate definitively incorrect concept predictions. 2) *Soft Constraints* capture probabilistic clinical knowledge. Unlike hard constraints, these represent strong statistical tendencies rather than absolute certainties. A deviation from these soft constraints does not necessarily indicate an error but flags a prediction as clinically less likely. To operationalize these patterns, we derive two key priors from the DRL6k:

$$\mathbf{P} \in \mathbb{R}^{(G+1) \times K}, \quad [\mathbf{P}]_{g,k} = P(c_k = 1 | y = g), \tag{2}$$

$$\mathbf{C} \in \mathbb{R}^{K \times K}, \quad [\mathbf{C}]_{i,j} = P(c_j = 1 | c_i = 1). \tag{3}$$

Specifically, $\mathbf{P}$ models the lesion profile conditioned on disease label, and $\mathbf{C}$ captures the statistical co-occurrence relationships between lesion types. We confirmed that these data-driven priors align well with established clinical knowledge from authoritative medical sources (American Academy of Ophthalmology, 2025), which validates their reliability for our framework. Furthermore, it is important to note that since $\mathbf{P}$ and $\mathbf{C}$ are derived exclusively from label correlations, they function independently of the pixel-level data; consequently, these priors are unaffected by variations in device protocols or potential ethical biases inherent to the raw image data.

### 3.3 KNOWLEDGE CONSTRAINED DISTILLATION

**Self Correction.** When $t$ violates hard constraints, we perform direct corrections to ensure clinical consistency. For healthy cases with detected lesions, we zero out all lesion probabilities. For diseased cases with no detected lesions, we generate corrected probabilities by sampling from the $P$ and $C$. This process can be formally defined as:

$$t' = \begin{cases} \mathbf{0} & \text{if } y = 0 \wedge \max(t) > \tau, \\ \mathcal{S}([\mathbf{P}]_y, \mathbf{C}) & \text{if } y > 0 \wedge \max(t) \leq \tau, \\ t & \text{otherwise}, \end{cases} \tag{4}$$

$$\mathcal{S}([\mathbf{P}]_y, \mathbf{C}) = \max(\tilde{t}, \ (\tau + \epsilon) \cdot E), \tag{5}$$

$$E = \left((\mathbf{C} > \tau)^{\top}(\tilde{t} > \tau) > 0\right) \odot (\tilde{t} \leq \tau), \tag{6}$$

where $\tau = 0.5$ is a natural decision boundary for lesion detection, and $\epsilon$ controls the activation strength of co-occurring lesions. This process begins by sampling $\tilde{t}$ from $[\mathbf{P}]_y$ to satisfy the basic correspondence between lesions and disease grades. Then, for activated lesions, we promote the activation of other lesions with strong co-occurrence according to $\mathbf{C}$, yielding the final corrected output $t'$. This ensures that the guidance from $T$ are consistent with established medical knowledge.

**Distillation Reweighting.** For samples that deviate from soft constraints, we cannot definitively label them as incorrect predictions. Instead, we design $\mathcal{L}_{soft}$ to quantify the degree of deviation from statistical priors and use it to adaptively reweight their influence during knowledge distillation. Specifically, we define it as:

$$\mathcal{L}_{soft}(t', y) = \underbrace{\sum_{k \in A_a \Delta A_e} |\tau - t'_k|}_{\text{Deviation from } \mathbf{P}} + \underbrace{\sum_{k \in A_c \setminus A_a} (\tau - t'_k)}_{\text{Deviation from } \mathbf{C}}, \tag{7}$$

where $A_a = \{k \mid t'_k \geq \tau\}$ denotes the set of activated lesions, $A_e = \{k \mid P_{y,k} \geq \tau\}$ represents lesions expected to be active for disease grade $y$, and $A_c = \{k \mid \exists j \in A_a \text{ s.t. } [\mathbf{C}]_{j,k} \geq \tau, j \neq k\}$ captures lesions that should co-occur with the activated ones. The first term penalizes deviations from expected lesion activations, while the second term penalizes the absence of expected co-occurring lesions. A larger $\mathcal{L}_{soft}$ indicates greater deviation from medical priors, suggesting the prediction

may be less reliable. We therefore down-weight such samples by mapping $\mathcal{L}_{soft}$ into a confidence weight $\mathbf{w} \in (0, 1]$ via the exponential function:

$$\mathbf{w} = \exp(-\mathcal{L}_{soft}(t', y)). \tag{8}$$

**Refined Distillation Loss.** The final knowledge distillation loss incorporates the corrected $t'$ and adaptive weights $\mathbf{w}$:

$$\mathcal{L}_{KD} = \mathbf{w} \cdot \mathcal{L}_{BCE}(s, t'), \tag{9}$$

where $s$ is the predicted probabilities from CBM's concept layer, and $\mathcal{L}_{BCE}$ is the binary cross-entropy loss as defined in Equation 1. We actually add a temperature parameter to smooth the logits, but we omit it here for simplicity.

### 3.4 KNOWLEDGE CONSTRAINED REGULARIZATION

While refined distillation guides the CBM toward clinically plausible predictions by demonstrating what correct concept activations should look like, we complement this positive guidance with explicit constraints that directly penalize implausible behaviors. To achieve this, we introduce a direct regularization term $\mathcal{L}_{Reg}$ that combines violations of both hard and soft constraints, acting as a complementary mechanism to distillation:

$$\mathcal{L}_{Hard}(z, y) = \begin{cases} \sum_{k=1}^{K} \text{ReLU}(z_k) & \text{if } y = 0, \\ \text{ReLU}(- \max(z)) & \text{if } y > 0, \end{cases} \tag{10}$$

$$\mathcal{L}_{Reg} = \mathcal{L}_{Hard}(z, y) + \mathcal{L}_{Soft}(s, y), \tag{11}$$

where $z$ and $s$ are concept logits and probabilities from the CBM, and $\mathcal{L}_{Soft}$ refers to Equation (7). The ReLU activation ensures penalties are applied only when constraints are violated, creating a piecewise linear loss that is zero when medical rules are satisfied. For healthy cases, summing all positive logits encourages no concepts being activated, while for diseased cases, penalizing the negative maximum logit ensures at least one concept remains activated.

### 3.5 OVERALL TRAINING OBJECTIVE

Finally, we integrate all components and the primary classification task into a unified training objective, allowing the model to optimize both concept learning under knowledge constraints and the main classification task:

$$\mathcal{L}_{CE} = - \sum_{i=0}^{G} y_i \log \hat{y}_i, \tag{12}$$

$$\mathcal{L}_{total} = \mathcal{L}_{CE} + \mathcal{L}_{KD} + \mathcal{L}_{Reg}, \tag{13}$$

where $y_i$ and $\hat{y}_i$ are the ground-truth and predicted probabilities for grade $i$, respectively.

## 4 EXPERIMENTS

### 4.1 EXPERIMENT SETUP

**Benchmark.** Following the ESDG test in GDRBench (Che et al., 2023), we utilize DDR (Li et al., 2019) and EyePACS (Foundation, 2015) for model training and validation. We then evaluate its generalization on six unseen domains: DeepDR (Liu et al., 2022), Messidor (Abràmoff et al., 2016), IDRID (Porwal et al., 2018), APTOS (Karthik et al., 2019), FGADR (Zhou et al., 2021), and RLDR (Wei et al., 2021). In all these datasets, DR is graded into five levels: Non-DR, Mild, Moderate, Severe, and Proliferative. Following the convention in the benchmark, we use the Area Under the ROC Curve (AUC) and the F1-score as our primary metrics.

**Implementation Details.** We evaluate KCCL on two representative concept-based models: the Concept Embedding Model (CEM) (Espinosa Zarlenga et al., 2022) and CLAT (Wen et al., 2025). CEM is a versatile CBM architecture recognized for its balance between interpretability and accuracy, while CLAT is a specialized CBM designed for ophthalmology applications. For the CEM

Table 2: Performance of proposed KCCL and other existing methods on six unseen domains. KCCL$_{CEM}$ and KCCL$_{CLAT}$ denote our framework applied to two representative CBMs, CEM and CLAT, respectively. Evaluation metrics include AUC and F1-score (%). The best results are highlighted in **bold**, and the second-best results are underlined. The backbone is ResNet50 unless otherwise noted, where † denotes ViT and ⋆ denotes VMamba.

| Method | APTOS | | DeepDR | | FGADR | | IDRID | | Messidor | | RLDR | | Average | |
|---|---|---|---|---|---|---|---|---|---|---|---|---|---|---|
| | AUC | F1 | AUC | F1 | AUC | F1 | AUC | F1 | AUC | F1 | AUC | F1 | AUC | F1 |
| *General Domain Generalization Methods* | | | | | | | | | | | | | | |
| Mixup | 65.5 | 30.2 | 70.7 | 33.3 | 58.8 | 7.4 | 70.2 | 32.6 | 71.5 | 32.6 | 72.9 | 27.0 | 68.3 | 27.2 |
| MixStyle | 62.0 | 25.0 | 53.3 | 14.6 | 51.0 | 7.9 | 53.0 | 19.4 | 51.4 | 33.1 | 53.5 | 6.4 | 54.0 | 17.7 |
| DDAIG | 67.4 | 31.6 | 73.2 | 29.7 | 59.9 | 5.5 | 70.2 | 33.4 | 73.5 | 35.6 | 74.4 | 23.5 | 69.8 | 26.6 |
| ATS | 68.8 | 32.4 | 72.7 | 33.5 | 60.3 | 5.7 | 69.1 | 30.6 | 73.4 | 32.4 | 75.0 | 23.9 | 69.9 | 26.4 |
| Fishr | 64.5 | 31.0 | 72.1 | 30.1 | 56.3 | 7.2 | 71.8 | 30.6 | 74.3 | 33.8 | 78.6 | 21.3 | 69.6 | 25.7 |
| MDLT | 67.6 | 32.4 | 73.1 | 33.7 | 57.1 | 7.8 | 71.9 s | 32.4 | 73.4 | 34.1 | 76.6 | 30.0 | 70.0 | 28.4 |
| *DR Grading Methods* | | | | | | | | | | | | | | |
| GREEN | 67.5 | 33.3 | 71.2 | 31.1 | 58.1 | 6.9 | 68.5 | 33.0 | 71.3 | 33.1 | 71.0 | 27.8 | 67.9 | 27.5 |
| CABNet | 67.3 | 30.8 | 70.0 | 32.0 | 57.1 | 7.5 | 67.4 | 31.7 | 72.3 | 35.3 | 75.2 | 25.4 | 68.2 | 27.1 |
| MIL-VT† | 69.1 | 36.8 | 78.3 | 36.3 | 62.1 | 9.3 | 71.7 | 31.1 | 78.3 | 40.7 | 80.8 | 34.5 | 73.4 | 31.5 |
| RETFound† | 81.2 | 41.4 | 78.2 | 31.1 | 77.9 | 34.9 | 85.6 | 45.8 | 81.9 | 43.5 | 81.1 | 40.9 | 81.0 | 39.6 |
| *Domain Generalizable DR Grading Methods* | | | | | | | | | | | | | | |
| DRGen | 69.4 | 35.7 | 78.5 | 31.6 | 59.8 | 8.4 | 70.8 | 30.6 | 77.0 | 37.4 | 78.9 | 21.2 | 72.4 | 27.5 |
| GDRNet | 69.8 | 35.2 | 76.1 | 35.0 | 63.7 | 9.2 | 72.9 | 35.1 | 78.1 | 40.5 | 79.7 | 37.9 | 73.4 | 32.2 |
| GAD† | 70.1 | 37.8 | 80.2 | 36.7 | 65.7 | 10.4 | 72.5 | 32.9 | 79.7 | 41.2 | 80.9 | 36.1 | 74.9 | 32.5 |
| DECO | 70.6 | 36.4 | 78.2 | 35.8 | 65.5 | 11.7 | 74.2 | 38.7 | 79.8 | 44.9 | 81.1 | 40.8 | 74.9 | 34.7 |
| Samba⋆ | / | 37.9 | **83.9** | **40.7** | / | 40.5 | / | 41.7 | / | 41.8 | / | 42.6 | / | 40.9 |
| *Concept-based Models for Generalization* | | | | | | | | | | | | | | |
| AlignCBM | 73.8 | 42.1 | 76.8 | 34.0 | 78.8 | 39.0 | 81.5 | 41.1 | 77.6 | 38.6 | 80.3 | 41.0 | 78.1 | 39.3 |
| KnoBo | 76.6 | 44.6 | 77.0 | 34.8 | 81.5 | 39.2 | 81.0 | 44.0 | 81.7 | 48.0 | 82.4 | 42.8 | 80.1 | 42.2 |
| KCCL$_{CEM}$ | 81.8 | 44.1 | 81.7 | 38.4 | **82.3** | **41.0** | 85.8 | 48.0 | **84.7** | **51.5** | 83.6 | **45.9** | 83.3 | 44.8 |
| KCCL$_{CLAT}$† | **82.2** | **47.8** | 83.9 | 40.2 | 81.3 | 40.6 | **89.3** | **52.6** | 83.9 | 51.2 | **84.2** | 44.3 | **84.1** | **46.1** |

implementation, we utilize a ResNet50 (He et al., 2016) backbone for consistency with the compared methods and set the concept embedding dimension to 512. For CLAT, we directly use the original configurations. All models are trained for up to 100 epochs using the AdamW optimizer with a batch size of 32. We set the initial learning rate to 5e-5, weight decay to 1e-4, activation strength parameter $\epsilon$ to 0.2, and the distillation temperature $\theta$ to 2.0. A CosineAnnealingWarmRestarts scheduler reduces the learning rate to a minimum of 1e-5. Early stopping is applied if validation loss does not improve for 10 epochs.

## 4.2 COMPARISON WITH STATE-OF-THE-ART

We compare KCCL with three categories of methods: 1) general domain generalization methods, including Mixup (Zhang et al., 2018), MixStyle (Zhou et al., 2020b), DDAIG (Zhou et al., 2020a), ATS (Yang et al., 2021), Fishr (Rame et al., 2022), MDLT (Yang et al., 2022); 2) DR grading methods, such as GREEN (Liu et al., 2020), CABNet (He et al., 2021), MIL-VT (Yu et al., 2021), and RETFound (Zhou et al., 2023); 3) domain generalizable DR grading methods, including DRGen (Atwany et al., 2022), GDRNet (Che et al., 2023), GAD (Bi et al., 2025), DECO (Xia et al., 2024), and Samba (Bi et al., 2024); 4) Concept-based models for generalization, including AlignCBM (Pang et al., 2024) and KnoBo (Yang et al., 2024). The results are cited from the original papers, except for AlignCBM and KnoBo, which we conduct the experiments based on their released codebases.

Table 2 presents the results of all methods across six target domains. Our KCCL framework achieves the best performance on nearly all target domains, demonstrating significant improvements over the state-of-the-art domain generalizable DR methods. The best-performing variant, KCCL$_{CLAT}$, exceeds the best baseline by 4.0% and 3.9% in AUC and F1-score. KCCL$_{CEM}$ also delivers strong performance, securing the second-best overall results. The improvements are particularly substantial on challenging domains such as IDRID and APTOS, where KCCL$_{CLAT}$ achieves remarkable F1-score improvements of 10.6% and 3.2%, respectively. Among the two variants, KCCL$_{CLAT}$ demonstrates superior performance, benefiting from its design tailored for fundus diagnosis, which provides in-

herent advantages for learning DR-related concepts. The remarkable performance improvement can be attributed to two key factors. First, the CBM architecture enables explicit concept-based reasoning, where lesion information is directly modeled to facilitate the model capturing more diagnostically meaningful features. Second, the knowledge constraint mechanism actively steers the concept learning process. Rather than simply using distillation to overcome annotation scarcity, KCCL imposes additional constraints derived from medical knowledge. This constraint fosters the learning of concept predictions that are well-aligned with domain-invariant medical principles. In turn, these medically-grounded concepts provide a more robust foundation for the final grading task, enhancing its accuracy and generalizability.

The consistent improvements across diverse datasets confirm that our KCCL framework enables robust domain generalization across different domains.

Table 3: Ablation study on each component of KCCL. The ID column serves as an index for easy reference to each experiment. KD: Knowledge Distillation, SC: Self Correction, DR: Distillation Reweighting, KCR: Knowledge Constrained Regularization. Grading refers to the averaged results on six unseen domains, while Lesion refers to the results on DRL6k test dataset. The best results are highlighted in **bold**, and the second-best results are underlined.

| ID | KD | SC | DR | KCR | CLAT Grading AUC | CLAT Grading F1 | CLAT Lesion AUC | CLAT Lesion F1 | CEM Grading AUC | CEM Grading F1 | CEM Lesion AUC | CEM Lesion F1 |
|---|---|---|---|---|---|---|---|---|---|---|---|---|
| 0 | ✗ | ✗ | ✗ | ✗ | 73.4 | 31.5 | / | / | 67.1 | 27.3 | / | / |
| 1 | ✗ | ✗ | ✗ | ✓ | 72.6 | 33.0 | 64.3 | 37.9 | 76.5 | 30.6 | 58.0 | 39.4 |
| 2 | ✓ | ✗ | ✗ | ✗ | 78.1 | 39.3 | 74.6 | 67.8 | 79.0 | 38.6 | 73.3 | 66.4 |
| 3 | ✓ | ✓ | ✗ | ✗ | 82.3 | 43.9 | 82.6 | 72.1 | 81.0 | 42.2 | 79.7 | 70.7 |
| 4 | ✓ | ✓ | ✗ | ✓ | 83.2 | 44.1 | 85.8 | 75.2 | 82.2 | 43.0 | 84.6 | 76.2 |
| 5 | ✓ | ✗ | ✓ | ✗ | 82.1 | 42.2 | 79.2 | 71.5 | 78.1 | 39.3 | 77.4 | 68.8 |
| 6 | ✓ | ✗ | ✓ | ✓ | 81.8 | 43.9 | **87.2** | 74.2 | 82.4 | 42.2 | 83.9 | 74.3 |
| 7 | ✓ | ✓ | ✓ | ✗ | 84.0 | 45.2 | 84.2 | 73.2 | **83.3** | 43.8 | 82.6 | 72.7 |
| 8 | ✓ | ✓ | ✓ | ✓ | **84.1** | **46.1** | 87.0 | **76.3** | **83.3** | **44.8** | **86.5** | **77.0** |

## 4.3 ABLATION STUDIES

To rigorously assess the contribution of each component within our KCCL framework, we conduct comprehensive ablation studies. Table 3 presents the results across different component combinations for both CLAT and CEM settings, evaluated on both DR grading (averaged across six unseen domains) and lesion detection (on DRL6k test set) tasks. Furthermore, to provide a more comprehensive analysis, we conducted additional experiments examining the impact of random priors, different $\epsilon$ values, and CBMs training on DRL6k only (see Appendix).

**Effect of Concept-based Reasoning.** We first evaluate the fundamental benefit of incorporating a concept layer trained via knowledge distillation (KD). Comparing the baseline model (ID 0) with a CBM trained only with standard KD (ID 2), we observe a substantial performance leap on the grading task: the F1-score increases from 31.5% to 39.3% for CLAT and from 27.3% to 38.6% for CEM. This demonstrates that explicitly modeling lesion concepts as an intermediate reasoning layer significantly enhances the model's ability to capture disease-relevant patterns. The learned lesion concepts provide discriminative features that directly facilitate grading decisions, as the model can leverage specific pathological indicators rather than relying on low-level visual features alone. In contrast, applying only the knowledge constrained regularization (KCR) without any concept supervision from KD (ID 1) yields limited improvement, underscoring the necessity of KD for providing the primary guidance for concept learning.

**Dissecting the Knowledge Constrained Distillation.** We then evaluate the impact of our knowledge constrained distillation, which is designed to provide reliable concept supervision by guiding the model on "what-to-learn." Self-correction (SC) rectifies distillation targets that violate hard constraints, while distillation reweighting (DR) dynamically adjusts the loss based on deviations from soft constraints. The two components individually lead to performance gains over plain distillation (ID 3 vs. ID 2, ID 5 vs. ID 2), with the correction mechanism showing larger improvements. This is likely because correcting implausible concept predictions particularly helps in identifying Non DR cases, which constitute a substantial proportion of the evaluation data. The reweighting component

provides smaller but also consistent gains (ID 3 vs. ID 5) by reducing the influence of uncertain concept predictions. When both correction and reweighting are combined (ID 7), we observe the largest improvements, further confirming that these two mechanisms complement each other effectively.

**The Role of Knowledge Constrained Regularization.** The knowledge constrained regularization (KCR) module enforces clinical consistency by penalizing implausible concept predictions. However, when used in isolation, KCR is insufficient to support effective concept learning (ID 1). Furthermore, our experimental results demonstrate that KCR exerts a relatively modest impact on final grading performance (*e.g.*, ID 8 vs. ID 7, ID 4 vs. ID 3). Nevertheless, KCR's primary contribution lies in enhancing the quality of the learned concepts themselves. As detailed in Table 3, KCR consistently boosts concept prediction metrics (*e.g.*, CLAT AUC improves from 84.2% to 87.0% in ID 8 vs. ID 7). This demonstrates that KCR is critical for producing more reliable and clinically coherent concepts, which is essential for model interpretability and trustworthiness.

**Sensitivity to Knowledge Distillation Temperature.** We further examine the impact of distillation temperature $\theta$ on KCCL's performance (Figure 2). The analysis reveals an optimal range for this hyperparameter. Performance remains robust across $\theta \in [1, 4]$, but deteriorates at higher values. This degradation occurs because excessive temperature smoothing erases the nuanced patterns that distinguish between different lesion types—precisely the fine-grained knowledge our framework aims to distill. We therefore set $\theta = 2.0$, which provides an optimal trade-off between preserving teacher model specificity and enabling stable concept learning.

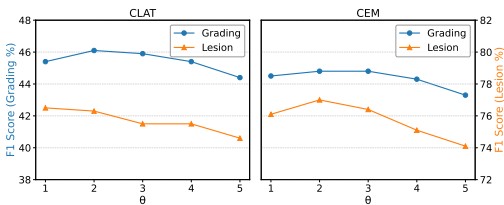 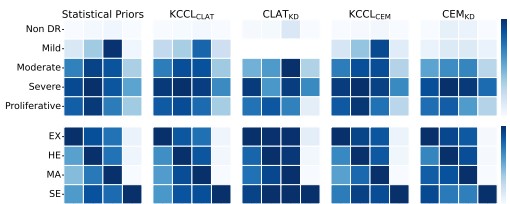

Figure 2: Effect of knowledge distillation temperature $\theta$ on the performance of CLAT and CEM on six unseen domains for grading and on DRL6k test dataset for lesion detection, where both models are trained with the proposed KCCL framework.

Figure 3: Heatmap visualization of **P** and **C**, along with the correlation between concept and grading predictions for CLAT and CEM on the DRL6k test set. Here, KCCL$_{CLAT}$ and KCCL$_{CEM}$ denote models trained with the proposed KCCL method, while CLAT$_{KD}$ and CEM$_{KD}$ use knowledge distillation only.

### 4.4 INTERPRETABILITY ANALYSIS

To evaluate how KCCL improves the consistency between concept predictions and decision-making, we visualize two key aspects and compare them against statistical priors (Figure 3): the association between predicted lesion concepts and the final DR grade, and the co-occurrence patterns among the concepts themselves. The heatmaps reveal that models trained with KCCL (KCCL$_{CLAT}$, KCCL$_{CEM}$) learn concept relationships that closely align with the medical priors. In contrast, baseline models trained only with knowledge distillation (CLAT$_{KD}$, CEM$_{KD}$) exhibit clinically implausible behaviors. For instance, CLAT$_{KD}$ incorrectly associates Non DR predictions with MA and fails to activate any relevant concepts for Mild DR, which is a clear contradiction of medical knowledge. Furthermore, both baseline models learn high correlations between all lesions, a pattern inconsistent with clinical reality. These findings demonstrate that KCCL effectively corrects the model's internal reasoning. By enforcing medical constraints, it produces concept predictions that are more faithful and aligned with medical expertise, which is a critical step toward building a truly generalizable model. We also provide case studies in Appendix to further illustrate the interpretability of our method.

## 5 DISCUSSION & CONCLUSION

Our proposed KCCL framework demonstrates significant promise in addressing domain shift challenges for DR grading through concept-based learning. By leveraging knowledge distillation from the lesion detection model and incorporating medical knowledge constraints, KCCL achieves su-

perior generalization performance while maintaining clinical interpretability. The extensive experiments across multiple unseen domains validate the effectiveness of our approach, consistently outperforming state-of-the-art methods. Despite these promising results, it is crucial to discuss the inherent limitations of our approach. The framework's performance, while robust, is fundamentally linked to the quality and scale of the initial knowledge source. Although KCCL alleviates the need for direct concept annotations, it introduces a dependency on a well-annotated lesion detection dataset to provide the initial supervision; however, such supervision is arguably necessary in the medical domain to ensure precise alignment between features and clinical semantics. Moreover, the statistical priors employed as knowledge constraints, while carefully validated, inherently represent common clinical patterns. They may not fully capture atypical disease presentations or rare lesion co-occurrence relationships that deviate from statistical norms. Furthermore, our current concept vocabulary is not exhaustive; the focus on four primary lesions omits critical PDR indicators like Neovascularization (NV), which may affect diagnostic accuracy for advanced grades. It should be noted that this limitation arises from the constraints of existing datasets rather than the KCCL framework itself. And the assumption of a complete concept vocabulary is inherently idealized, as even clinical practice continues to refine diagnostic criteria. Nonetheless, the KCCL framework is inherently flexible and can be extended to accommodate additional concepts as they become available.

In conclusion, KCCL offers a promising approach toward practical, interpretable, and generalizable medical AI. It provides a principled approach to learning robust concept representations under limited supervision, a challenge common to many medical imaging tasks. Future work should focus on further reducing the dependency on annotated data through semi-supervised or self-supervised paradigms, expanding the concept vocabulary to cover a wider range of pathologies, and exploring methods to dynamically discover and adapt clinical constraints from data.

## ETHICS STATEMENT

All experiments in this study were conducted on publicly available datasets. The DRL6k dataset, which we curated and utilized, was aggregated and processed entirely from these existing public sources. Our research did not involve any new data collection, clinical trials, or human subjects. The study exclusively used anonymized data, raising no new privacy or security concerns. It is crucial to note that our model is developed for research purposes only and is not intended for direct clinical diagnosis or as a substitute for evaluation by a qualified medical professional. While important considerations for clinical deployment, topics such as algorithmic bias and fairness are outside the scope of this paper.

## REPRODUCIBILITY STATEMENT

All details regarding our proposed KCCL framework, including the design of knowledge constraints, self correction mechanism, distillation reweighting, knowledge constrained regularization, are comprehensively described in Section 3. The hyperparameter settings and experimental configurations are provided in Section 4.1 and Appendix. The source code will be made publicly available upon acceptance.

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

# A  APPENDIX

**The Impact of Medical Priors.** To further investigate the impact of medical priors on our proposed method, we replaced the established priors with randomly generated priors during training. Based on the results of multiple experiments, we plotted a bar chart as shown in Figure 4. The results demonstrate that, overall, the performance with random priors is significantly worse. This is likely because random priors are unable to provide effective guidance and, in many cases, may mislead the model into learning incorrect conceptual relationships. Consequently, this negatively impacts the overall performance of the model, resulting in significant variability in the outcomes, both in terms of grading and lesion detection.

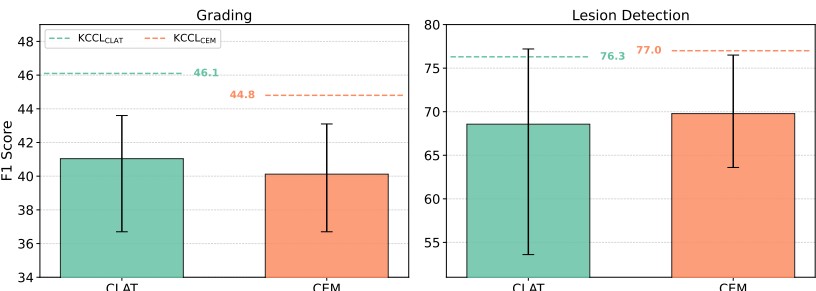

Figure 4:  Performance comparison between using true medical priors and random priors in KCCL$_{\mathrm{CLAT}}$ and KCCL$_{\mathrm{CEM}}$. The bars represent the mean F1-scores averaged over five independent runs with different random seeds, with error bars indicating the range of variation. The dashed lines represent the performance achieved using true medical priors for KCCL$_{\mathrm{CLAT}}$ (green) and KCCL$_{\mathrm{CEM}}$ (orange), respectively.

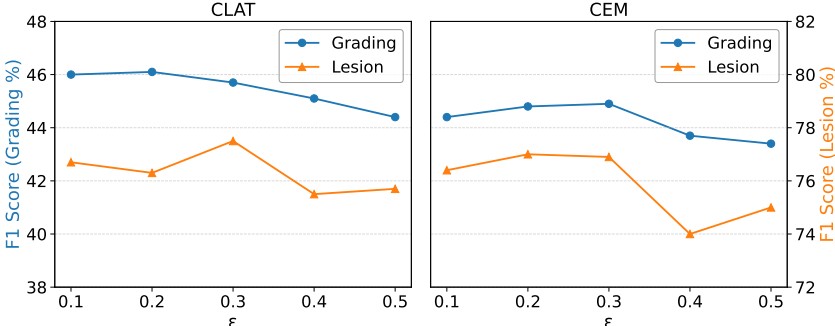

Figure 5:  Effect of co-occurrence activation strength $\epsilon$ on the grading performance of KCCL$_{\mathrm{CLAT}}$ and KCCL$_{\mathrm{CEM}}$ on six unseen domains and the lesion detection performance on DRL6k test dataset.

**Sensitivity to Co-occurrence Activation Strength.** To analyze the impact of co-occurrence activation strength parameter $\epsilon$ on our method, we further examine the effect of varying this parameter, as shown in Figure 5. The results indicate that our method is generally robust to the choice of $\epsilon$, achieving good performance within the range of 0.1 to 0.3. However, when $\epsilon$ is set too high, related lesions may become overly activated, and the sampled lesion probabilities can be excessively influenced. This leads to a corrected probability distribution that deviates significantly from the true distribution, thereby adversely affecting the learning of the concept layer and ultimately impacting grading performance to some extent. Therefore, we selected $\epsilon = 0.2$ for our experiments as it provides a more balanced training signal.

**Sensitivity to Lesion Detection Threshold.** Our selection of $\tau = 0.5$ follows the standard convention for binary classification logits. To address the question of sensitivity, we performed an ablation study ranging from 0.1 to 0.9, as shown in Figure 6. As the results show, the model's performance peaks in the 0.4-0.5 range. KCCL is designed to be robust to teacher imperfections and does not

rely on a perfect lesion detector. Consequently, extensive tuning of the threshold is not required. Furthermore, we avoided using adaptive or learnable threshold mechanisms, as optimizing these parameters on the source domain poses a risk of overfitting.

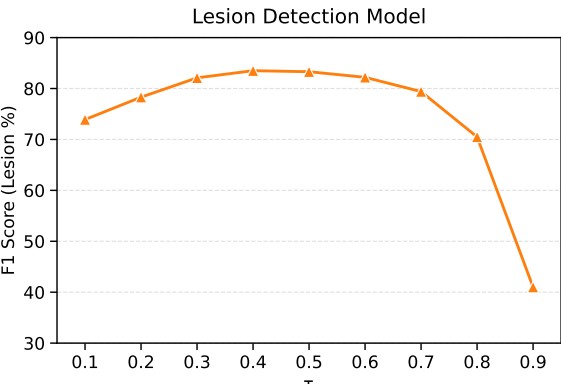

Figure 6: Impact of lesion detection threshold $\tau$.

Table 4: Performance of concept-based models trained only on the DRL6k dataset. Here, $\text{KCCL}_\text{CLAT}$ and $\text{KCCL}_\text{CEM}$ denote models trained with the proposed KCCL method, while $\text{CLAT}_\text{KD}$ and $\text{CEM}_\text{KD}$ use knowledge distillation only.

| Method | Grading | | Lesion | |
|---|---|---|---|---|
| | AUC | F1 | AUC | F1 |
| $\text{CEM}_\text{DRL6k}$ | 79.5 | 36.8 | 89.3 | 79.7 |
| $\text{KCCL}_\text{CEM}$ | $83.3_{\uparrow 3.8}$ | $44.8_{\uparrow 8.0}$ | $86.5_{\downarrow 2.8}$ | $77.0_{\downarrow 2.7}$ |
| $\text{CLAT}_\text{DRL6k}$ | 80.3 | 37.6 | 89.6 | 80.0 |
| $\text{KCCL}_\text{CLAT}$ | $84.1_{\uparrow 3.8}$ | $46.1_{\uparrow 8.5}$ | $87.0_{\downarrow 2.6}$ | $76.3_{\downarrow 3.7}$ |

**Concept-based models with Only DRL6k Supervision.** Some studies (Koh et al., 2020; Espinosa Zarlenga et al., 2022) have shown that concept-based models are inherently more data-efficient. To further illustrate the effectiveness of the additional data introduced through knowledge distillation, we trained CEM and CLAT solely on the DRL6k dataset and evaluated their generalization capabilities using the same benchmark (excluding FGADR due to its partial overlap with the training data). As shown in Table 4, CEM and CLAT trained only on DRL6k still exhibit a performance gap compared to KCCL. However, they already outperform many other domain generalization methods, highlighting the inherent generalization ability and data efficiency of concept-based models. Building on this foundation, KCCL further leverages the strengths of concept-based models by employing knowledge-guided training to enable concept learning without direct concept annotations. The incorporation of additional data significantly enhances performance on unseen domains, while only causing a slight decrease in lesion prediction performance. This greatly improves the practicality of concept-based models in medical imaging and offers a novel approach for enhancing model generalization in this field.

**Case Study.** Figure 7 presents representative cases that demonstrate KCCL's impact on interpretability and clinical consistency. The first two cases show case the ability of KCCL to distinguish Non DR from other grades. While $\text{CLAT}_\text{KD}$ frequently misclassifies these cases due to incorrect MA predictions, $\text{KCCL}_\text{CLAT}$ correctly identifies the absence of lesions in Non DR cases, leading to more accurate grading. The third case further illustrates KCCL's advantage in clinical consistency. Although both $\text{KCCL}_\text{CLAT}$ and $\text{CLAT}_\text{KD}$ predict a Proliferative outcome, their reasoning differs: $\text{CLAT}_\text{KD}$ shows inconsistent lesion activations that contradict the severity of the predicted grade, while $\text{KCCL}_\text{CLAT}$ demonstrates coherent lesion predictions that align with the medical expectations. This consistency is crucial for building clinician trust, as it ensures that the model's explanations faithfully reflect genuine clinical reasoning rather than spurious correlations.

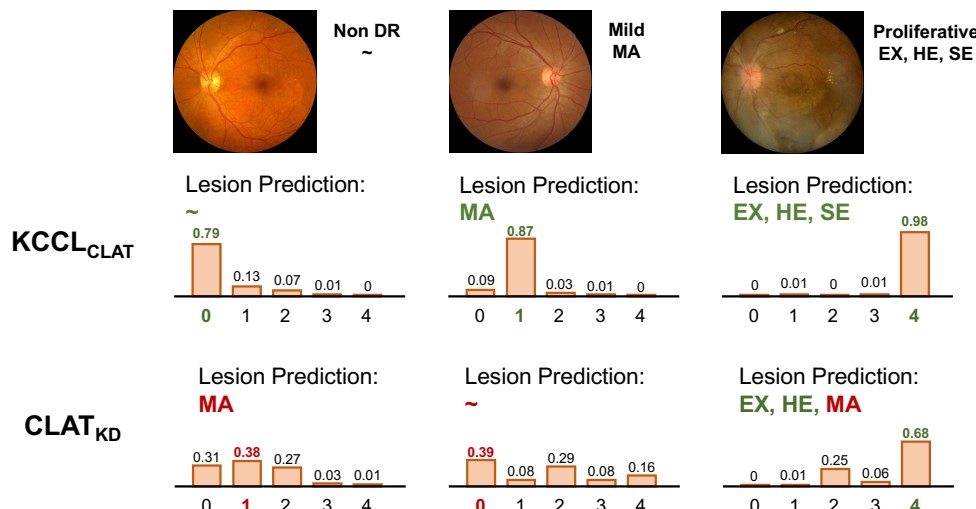

Figure 7: Case study of KCCL on DR grading and lesion detection. The first row shows the original images, with ground-truth DR grades and lesions underneath. The following rows show the predicted DR grades and lesions from $KCCL_{CLAT}$ and $CLAT_{KD}$. Grade predictions are shown as confidence score bar charts for the five DR grades (0: Non-DR, 1: Mild, 2: Moderate, 3: Severe, 4: Proliferative). The notation "$\sim$" indicates no lesion activation.

Table 5: Performance of different lesion detection models on the DRL6k test dataset.

| Model | Architecture | AUC | F1 |
|---|---|---|---|
| FLAIR | RN50 | 91.3 | 83.3 |
| RET-CLIP | ViT-B | 90.0 | 81.0 |
| ViLReF | ViT-B | 88.4 | 78.3 |
| RETFound | ViT-B | 91.7 | 82.9 |

Table 6: Impact of different teacher models.

| Method | FLAIR | | RET-CLIP | | ViLReF | | RETFound | |
|---|---|---|---|---|---|---|---|---|
| | AUC | F1 | AUC | F1 | AUC | F1 | AUC | F1 |
| $KCCL_{CLAT}$ | 84.2 | 46.0 | 84.1 | 45.9 | 81.3 | 46.3 | 80.7 | 45.3 |
| $KCCL_{CEM}$ | 83.0 | 44.7 | 83.0 | 44.8 | 82.9 | 44.2 | 83.1 | 43.5 |

**Lesion Detection Model Training.** To obtain the concept pseudo-labels required for knowledge distillation in KCCL, we trained a lesion detection model to serve as the teacher model. This model is based on a ResNet50 backbone, initialized with pre-trained weights from the image encoder of FLAIR (Silva-Rodríguez et al., 2025). The model architecture remains unaltered, employing only CutMix data augmentation to enhance its generalization capabilities, with the CutMix alpha parameter set to 1.0. Additionally, reweighting is applied to address class imbalance, where the weights are set as the inverse of the sample proportion for each lesion category. Consistent with the main experiments of KCCL, the training utilizes the AdamW optimizer with a learning rate of 1e-4, weight decay of 1e-4, batch size of 32, and a maximum of 100 training epochs. In addition to FLAIR, we also trained other models including RET-CLIP (Du et al., 2024), ViLReF (Yang et al., 2025), and RETFound Zhou et al. (2023) following a similar protocol to evaluate different teacher architectures. The performance of these models on the DRL6k test dataset is presented in Table 5. Furthermore, the impact of utilizing these different teacher models on the downstream performance of KCCL is presented in Table 6. This confirms that the effectiveness of our knowledge constraints is not tied to a specific architecture. Crucially, this stability reaffirms the core premise of our work: the teacher

model primarily serves to provide a basic supervision signal for concept learning, while the KCCL framework itself is the more critical component.

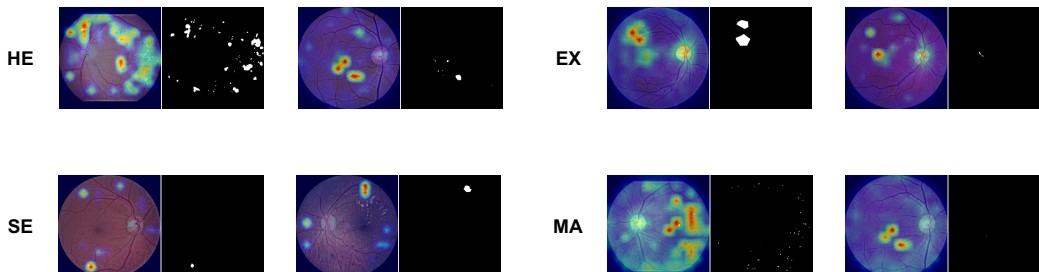

Figure 8: Visualization of KCCL$_{CLAT}$ for lesion localization.

**Visualization of Lesion Localization.** To verify the model's focus, we visualized the attention regions of KCCL$_{CLAT}$ regarding different lesion types on the DRL6k test dataset. As depicted in Figure 8, the results indicate that the concept layer captures and focuses on key lesion regions to a reasonable degree. Notably, even under the constraint of a purely pseudo-label training regime, the model still exhibits a discernible tendency to attend to clinically relevant areas, reinforcing the validity of the learned representations.

**Stability Analysis.** Although the original benchmark protocol primarily reports single-run performance without quantifying statistical dispersion, we conducted three independent runs to rigorously evaluate the stability of the KCCL framework. Table 7 presents the performance in terms of mean $\pm$ standard deviation. The results exhibit minimal variance across different initializations, confirming that our method maintains consistent and stable performance

Table 7: Domain generalization performance (Mean $\pm$ Standard Deviation) across multiple runs. The results are reported in terms of AUC (%) and F1 scores (%).

| Method | APTOS | | DeepDR | | FGADR | | IDRID | | Messidor | | RLDR | | Average | |
|---|---|---|---|---|---|---|---|---|---|---|---|---|---|---|
| | AUC | F1 | AUC | F1 | AUC | F1 | AUC | F1 | AUC | F1 | AUC | F1 | AUC | F1 |
| KCCL$_{CEM}$ | 81.4 ±0.4 | 44.3 ±0.7 | 81.0 ±0.7 | 38.8 ±1.2 | 81.8 ±1.6 | 40.7 ±2.0 | 84.7 ±0.9 | 47.5 ±0.7 | 84.9 ±0.3 | 51.1 ±0.4 | 84.0 ±0.3 | 46.1 ±0.3 | 83.0 ±0.4 | 44.7 ±0.6 |
| KCCL$_{CLAT}$ | 82.0 ±0.3 | 47.1 ±0.7 | 83.0 ±1.1 | 39.2 ±3.0 | 81.6 ±0.3 | 40.6 ±0.8 | 89.5 ±0.3 | 52.7 ±3.7 | 84.8 ±0.9 | 51.2 ±1.3 | 84.3 ±0.1 | 45.1 ±0.8 | 84.2 ±0.2 | 46.0 ±0.5 |

**The Use of Large Language Models.** We used large language models (LLMs) solely for language polishing purposes, such as improving grammar, readability, and clarity of writing. LLMs were not involved in research ideation, experimental design, analysis, or interpretation of results. All scientific contributions and substantive content of the paper are entirely the work of the authors.

