# OpenReview forum: "Generalizable Diabetic Retinopathy Grading via Knowledge Constrained Concept Learning"
_ICLR.cc/2026/Conference — ICLR 2026 Conference Withdrawn Submission_

### Official Review · Reviewer_2Pjq · 2025-10-31

**Soundness:** 2
**Presentation:** 3
**Contribution:** 2
**Rating:** 4
**Confidence:** 3

**Summary:**

This paper addresses the domain generalization problem in DR grading—a critical issue where models trained on one dataset fail to generalize across diverse clinical settings. The authors propose Knowledge Constrained Concept Learning (KCCL), a framework that integrates concept-based modeling, knowledge distillation, and medical knowledge constraints to achieve clinically consistent and generalizable predictions. The authors first curate a new `DRL6k` dataset, containing 6,000 fundus images annotated with four key lesion types (i.e. hard/soft exudates, microaneurysms, hemorrhages). They then train a lesion detection model as a teacher, and then constrain the student concept bottleneck model (CBM) via a knowledge-constrained distillation.

**Strengths:**

- The proposed method is timely and clinically meaningful. The paper tackles a major bottleneck in medical AI—domain generalization—with a clinically guided approach that aligns with human diagnostic reasoning.
- The integration of concept bottlenecks with knowledge-constrained distillation is interesting. The proposed three mechanisms (i.e. Self-Correction, Reweighting, Regularization) seamlessly integrate medical knowledge and balance robustness and interpretability.
- Evaluations span six unseen domains and include detailed ablations isolating each component’s contribution. The improvements are consistent and significant across both AUC and F1 metrics. The correlation heatmaps between learned lesion concepts and DR grades demonstrate clinically coherent reasoning.

**Weaknesses:**

- My biggest concern is the dependency on the curated lesion detection model. While DRL6k partially mitigates annotation scarcity, the overall performance still depends on the teacher model’s quality. The resting of the framework assumes access to a reasonably accurate lesion detector. Unfortunately, the curated DRL6k dataset appears to be quite small (~6,000 samples), which does not lead to a high accuracy of the lesion detection model (F1=83.3). The curation of the DRL6k dataset also relies on image-level labels.

- Following my previous point, with the proposed settings, it seems that all the methods do not achieve a high accuracy in DR grading (with the highest F1=47.8). But training solely on e.g., Eyepacs and evaluating on APTOS can easily achieve an F1 score of 80. Then what is the point of applying the proposed framework ?

- The use of only four lesion types in the curated dataset omits key proliferative indicators (e.g., neovascularization), potentially constraining the grading accuracy for advanced DR.

- Currently, the lesion detection model is a standard ResNet50. Can the authors try different teacher models like ViTs to see the effect of the teacher?

**Questions:**

- It appears that the proposed method is tailored to DR grading. Can the method be generalized to other tasks, such as lesion segmentation.
- The authors should provide some visualization of the lesion localization to verify if the model indeed focuses on lesion regions.
- While Figure 3 provides interpretability analysis, qualitative failure examples (e.g., false lesion activations under domain shift) would enhance transparency.

---

> ### Author Response · Authors · 2025-11-20
>
> We sincerely thank the reviewer for the constructive and insightful feedback. We appreciate your recognition of the clinical relevance of our work and the effectiveness of the proposed framework in tackling domain generalization. We have addressed each point below, supported by additional experiments and detailed clarifications.
>
> ### 1. Dependency on the curated lesion detection model
>
> We thank the reviewer for this insightful comment regarding the dependency on the lesion detection model. We acknowledge the concern that a teacher model trained on a limited dataset like DRL6k may not be perfect. In fact, this precise challenge was a primary motivation for developing our KCCL framework.
>
> **As discussed in Section 3.2, we recognize that a lesion detector trained on a small-scale dataset is prone to generalization issues and can provide noisy pseudo-labels.** Therefore, instead of relying on simple knowledge distillation which would naively propagate these errors, we designed KCCL to explicitly manage and mitigate this noise. Our framework is not predicated on having a perfect teacher; rather, it is designed to be robust to an imperfect one.
>
> The core contribution of our work lies in the three knowledge-constrained mechanisms: 1) **Self-Correction** uses hard constraints derived from medical priors to rectify demonstrably incorrect pseudo-labels from the teacher. 2) **Distillation Reweighting** leverages soft constraints to down-weight the influence of samples where the teacher's predictions are clinically less plausible. 3) **Knowledge Constrained Regularization** directly penalizes the student model for producing clinically inconsistent concept predictions.
>
> These mechanisms work in concert to refine the noisy supervision signal, ensuring the student CBM learns clinically coherent and robust concepts. The results from our ablation study (Table 3, ID 2 vs. ID 8) empirically validate this, showing that KCCL's components provide a substantial performance gain over a baseline that uses only standard knowledge distillation. This demonstrates that the strength of our framework lies in its ability to harness medical knowledge to overcome the limitations of a weakly supervised teacher, rather than its dependency on it.
>
> ### 2. Generalization Performance
>
> We appreciate the reviewer raising this important point about the overall performance. As the reviewer insightfully noted in the summary, domain generalization is a **critical issue in DR grading** and our proposed method is **timely and clinically meaningful.** The seemingly modest F1-scores reported in our study are a direct reflection of this profound challenge.
>
> Our experimental setup strictly follows the ESDG test in the challenging benchmark ESGDRBench, proposed by Che et al. [1], which is designed to evaluate generalization to completely unseen domains. **Critically, all prior state-of-the-art domain generalization methods evaluated under this exact same rigorous protocol report F1-scores in a similar range.** As shown in Table 2, established methods such as DECO, GAD, and Samba achieve F1-scores ranging from 34.7% to 40.9% on average across the six unseen domains.
>
> Moreover, in case the reviewer finds the results based on this single benchmark [1] alone not sufficiently convincing, we provide further corroboration from a recent study [2]. Despite employing self-supervised learning on 15 fundus datasets followed by fine-tuning on EyePACS, they reported an F1-score of 46.4% on the unseen APTOS dataset. Our method achieves a comparable or superior F1-score (47.8%) using only two source datasets.
>
> **Regarding the comment that an F1-score of 80% can be easily achieved, we were unable to find a publication reporting such results under a strict cross-domain transfer setting without target domain adaptation.** We would be very grateful if the reviewer could share the specific reference, as it would be a valuable addition to our understanding of the field.

---

> > ### Author Response · Authors · 2025-11-20
> >
> > ### 3. Limited Concept Vocabulary
> >
> > We acknowledge the absence of certain lesion types (e.g., Neovascularization) as a limitation dictated by the availability of annotations in current public datasets, as stated in our discussion section. However, it is crucial to emphasize that this is a constraint of the **data**, not the **KCCL framework** itself.
> >
> > Furthermore, the assumption of a **complete** concept vocabulary is itself idealized. Even in clinical practice, ophthalmology continues to refine diagnostic criteria, and it remains uncertain whether all pathologically relevant features have been fully identified. From a Domain Generalization perspective, the core question is not whether the concept set is exhaustive, but whether the model can learn robust representations and have stable performance across domains. Our experimental results provide strong evidence for this: KCCL achieves state-of-the-art generalization across six unseen domains. Moreover, our method is inherently extensible and can readily accommodate richer concept sets as more comprehensively annotated datasets become available, further enhancing its diagnostic granularity.
> >
> > ### 4. Different teacher models
> >
> > We thank the reviewer for this constructive suggestion. To verify the robustness of our framework, we conducted additional experiments using various architectures as the teacher model. The results, which we have added to the revised supplementary material, demonstrate that KCCL consistently improves performance regardless of the teacher backbone used. This confirms that the effectiveness of our knowledge constraints is not tied to a specific architecture. Crucially, this stability reaffirms the core premise of our work: the teacher model primarily serves to provide a basic supervision signal for concept learning, while the KCCL framework itself is the more critical component.
> >
> > |  Model   | Architecture | AUC  |  F1  |
> > | :------: | :----------: | :--: | :--: |
> > |  FLAIR   |     RN50     | 91.3 | 83.3 |
> > | RET-CLIP |    ViT-B     | 90.0 | 81.0 |
> > |  ViLReF  |    ViT-B     | 88.4 | 78.3 |
> > | RETFound |    ViT-B     | 91.7 | 82.9 |
> >
> > | Teacher  | CLAT+KCCL |      | CEM+KCCL |      |
> > | :------: | :-------: | :--: | :------: | :--: |
> > |          |    AUC    |  F1  |   AUC    |  F1  |
> > |  FLAIR   |   84.2    | 46.0 |   83.0   | 44.7 |
> > | RET-CLIP |   84.1    | 45.9 |   83.0   | 44.8 |
> > |  ViLReF  |   81.3    | 46.3 |   82.9   | 44.2 |
> > | RETFound |   80.7    | 45.3 |   83.1   | 43.5 |
> >
> > ### 5. Adapt to segmentation
> >
> > Thank you for this interesting question. Our work, as stated in the title and introduction, is specifically focused on addressing the domain generalization challenge in the context of DR grading. Lesion segmentation represents a fundamentally different task with distinct requirements: it demands precise spatial localization at the pixel level, whereas our concept-based framework operates at the image level to model lesion presence for grading decisions.
> >
> > While extending KCCL to segmentation tasks lies beyond the scope of the current work, we acknowledge the potential for such adaptation. The core principles of KCCL, leveraging medical knowledge to guide model learning under limited supervision, could potentially inform future research on domain-generalizable segmentation. However, this would require substantial modifications to accommodate spatial reasoning and pixel-wise predictions. We welcome any further discussion on this topic.
> >
> > ### 6. Lesion localization & failure examples
> >
> > In the revised manuscript, we have addressed the request for localization verification by **including detailed visualizations in Appendix Fig. 8**, which qualitatively confirm that our model effectively focuses on the discriminative lesion regions. Regarding the suggestion to display false lesion activations under domain shift, we wish to clarify a specific constraint of the benchmark utilized in this study: **the dataset does not provide lesion annotations for the unseen domains**, which prevents the evaluation of lesion activation accuracy under domain shift. However, we would like to **respectfully draw the reviewer's attention to the case study already provided in the Appendix of our original submission**, which includes representative failure examples.
> >
> > ref:
> >
> > [1] Che et al. Towards generalizable diabetic retinopathy grading in unseen domains. MICCAI 2023
> >
> > [2] Galappaththige et al. Generalizing to unseen domains in diabetic retinopathy classification. WACV 2024

---

### Official Review · Reviewer_6szW · 2025-11-01

**Soundness:** 3
**Presentation:** 2
**Contribution:** 2
**Rating:** 2
**Confidence:** 4

**Summary:**

The paper proposes a framework for diabetic retinopathy (DR) grading that aims to improve cross-domain generalization. The authors integrate concept-based models (CBMs) with medical knowledge constraints, leveraging lesion detection and knowledge distillation to ensure clinically consistent and interpretable feature learning.

They construct a new DRL6k dataset with 6,000 annotated fundus images and introduce three components, including Self-Correction (SC), Distillation Reweighting (DR), and Knowledge Constrained Regularization (KCR), which are used to refine concept learning under medical priors. Experiments across multiple unseen datasets reportedly show performance improvements compared to prior domain generalization methods.

**Strengths:**

(1).Interpretability emphasis
The visual analyses showing better alignment between learned lesion concepts and disease grades highlight some degree of explainability.

(2). Structured integration of medical priors
The proposed knowledge constraints (hard and soft) are well-formulated and embedded in the training objective, enhancing interpretability.

(3). Relatively comprehensive experiments
Multiple datasets and ablation studies are included to assess contributions from different modules.

(4). Clear motivation and clinical grounding
The work identifies the domain shift problem in DR grading and builds from the observation that ophthalmologists rely on lesion-level reasoning.

**Weaknesses:**

(1). While the authors frame their method as introducing a “knowledge-constrained concept learning” approach, the underlying paradigm largely replicates well-established lines of work: for instance, Integrating Clinical Knowledge into Concept Bottleneck Models aligns CBMs with clinical knowledge for improved out-of-distribution (OOD) generalization [1]; several label-free or post-hoc CBM methods reduce reliance on concept annotations and perform concept distillation [2, 3]. Hence, the methodological contribution appears incremental rather than fundamentally novel.

[1] W. Pang, X. Ke, S. Tsutsui, and B. Wen, Integrating Clinical Knowledge into Concept Bottleneck Models, in Proc. MICCAI 2024.

[2] T. Oikarinen, S. Das, L. M. Nguyen, and T.-W. Weng, Label-Free Concept Bottleneck Models, ICLR 2023.

[3] D. Srivastava, G. Yan, and T.-W. Weng, VLG-CBM: Training Concept Bottleneck Models with Vision–Language Guidance, NeurIPS 2024.

(2). Underwhelming performance
The reported results are significantly lower than recent related paper [1], even simpler CNN-based networks, such as MobileNet and ResNet50, combined with data augmentation and regularization, can likely achieve comparable or better results[2,3].

[1].Che, Haoxuan, et al. "Towards generalizable diabetic retinopathy grading in unseen domains." International Conference on Medical Image Computing and Computer-Assisted Intervention. Cham: Springer Nature Switzerland, 2023.

[2].Zhu, Wenhui, et al. "nnMobileNet: rethinking CNN for retinopathy research." Proceedings of the IEEE/CVF Conference on Computer Vision and Pattern Recognition. 2024.

[3].Huang, Yijin, et al. "Identifying the key components in ResNet-50 for diabetic retinopathy grading from fundus images: a systematic investigation." Diagnostics 13.10 (2023): 1664.


(3). The curated DRL6k dataset is small (6k images) and includes only four lesion types (EX, SE, MA, HE), omitting key indicators like neovascularization (NV) that are crucial for proliferative DR. Complex dataset like EyePACS (Hard to identify grading 0 and 1) is used only for training rather than for evaluation, making the claim of generalization less convincing[1].

[1] Sun, Rui, et al. "Lesion-aware transformers for diabetic retinopathy grading." Proceedings of the IEEE/CVF Conference on Computer Vision and Pattern Recognition. 2021.

(4). The paper does not compare against recent retinal foundation models such as RetFound [1], which already show strong cross-domain generalization in retinal and medical imaging tasks without explicit domain adaptation.

[1] Li et al., RetFound: Foundation Model for Retinal Disease Detection, Nature, 2023.

**Questions:**

While the motivation and interpretability goals are valuable, the overall contribution is incremental and empirically weak. A lot of recent literature should be included.

---

> ### Author Response · Authors · 2025-11-20
>
> We sincerely thank the reviewer for their thorough evaluation and constructive feedback. We greatly appreciate the recognition of our work's strengths, including the interpretability emphasis, structured integration of medical priors, comprehensive experiments, and clear clinical grounding.
>
> We believe, however, that there may be some misunderstandings regarding the novelty of our methodological contribution and the rigor of our experimental evaluation. Below, we provide detailed clarifications and new experimental evidence to address each concern.
>
> ### 1. Methodological Contribution
>
> We thank the reviewer for their insightful feedback on the methodological contribution of our work. We wish to respectfully clarify a potential misunderstanding regarding the comparison to recent CBM literature.
>
> First, regarding AlignCBM [1], we would like to **note that this work is actually included as one of our baseline methods and discussed in our original manuscript**. While we acknowledge the shared high-level goal of integrating clinical knowledge, our approach is fundamentally different in both motivation and implementation. AlignCBM requires additional, fine-grained expert annotations (i.e., concept importance levels) for each sample, thereby increasing the annotation burden. In contrast, our KCCL framework is designed specifically to **alleviate the dependency on concept labels** for the main training data. KCCL leverages medical priors derived from a separate, smaller annotated dataset to guide concept learning on large, unlabeled datasets, with no need for clinician involvement during the main training phase. Our goal is to enhance scalability and practical applicability in label-scarce medical scenarios, a distinct motivation from AlignCBM.
>
> Second, concerning label-free CBMs [2, 3], we agree they represent a relevant line of work and have cited and discussed them in our initial manuscript. Their primary limitation, however, is a heavy reliance on the cross-modal capabilities of general-domain vision-language models (e.g., CLIP), whose performance is often suboptimal in specialized medical domains with unique concepts. To provide empirical evidence, **we have conducted new experiments with LF-CBM and VLG-CBM** on the benchmark. As shown in the new Table R1 (rows #0-1).
>
> ### 2. Underwhelming Performance
>
> We sincerely appreciate the reviewer's thoughtful feedback on our model's performance. We would like to provide additional context that may help clarify the experimental setup and evaluation protocol.
>
> First and foremost, we wish to clarify that our work utilizes the **exact same benchmark protocol as the primary reference cited by the reviewer**, Che et al. [4], specifically the **ESDG test** in GDRBench. Crucially, most of the baseline results reported in our paper are directly cited from this established benchmark. To avoid similar confusion, we have further clarified this in the revised manuscript. We hope this clarification resolves the concern that "the reported results are significantly lower than recent related paper[4]."
>
> Regarding the other cited works, nnMobileNet [5] and Huang et al. [6], they are valuable **but do not address the problem of domain generalization**. The work in [5] reports 5-fold cross-validation results on APTOS and binary classification results on other datasets, while [6] focuses on an ablation study of common training modules. Their experimental protocols are not designed to evaluate generalization to unseen domains and are thus not directly comparable to our DG setting. Nevertheless, to further substantiate our performance, **we have included nnMobileNet [5] in our experiments** within the ESDG test using the GDRBench protocol,  as shown in Table R1 (row #2).

---

> > ### Author Response · Authors · 2025-11-20
> >
> > ### 3. Limited Concept Vocabulary & EyePACS
> >
> > We acknowledge the absence of certain lesion types (e.g., Neovascularization) as a limitation dictated by the availability of annotations in current public datasets, as stated in our discussion section. However, it is crucial to emphasize that this is a constraint of the **data**, not the **KCCL framework** itself.
> >
> > Furthermore, the assumption of a **complete** concept vocabulary is itself idealized. Even in clinical practice, ophthalmology continues to refine diagnostic criteria, and it remains uncertain whether all pathologically relevant features have been fully identified. From a Domain Generalization perspective, the core question is not whether the concept set is exhaustive, but whether the model can learn robust representations and have stable performance across domains. Our experimental results provide strong evidence for this: KCCL achieves state-of-the-art generalization across six unseen domains. Moreover, our method is inherently extensible and can readily accommodate richer concept sets as more comprehensively annotated datasets become available, further enhancing its diagnostic granularity.
> >
> > Regarding the use of the EyePACS dataset for training rather than evaluation, this decision was made to strictly follow the established protocol of the GDRBench[4]. This adherence ensures that our results are directly and fairly comparable to other state-of-the-art domain generalization methods evaluated on the same benchmark. We believe that maintaining consistency with established evaluation standards is crucial for scientific rigor.
> >
> > ### 4. Foundation Models
> >
> > We thank the reviewer for the excellent suggestion to compare our method against recent retinal foundation models. Following this advice, we have now conducted a new experiment including RETFound [7]. The results are now included in our new Table R1 (row #3). Notably, because our fine-tuning dataset is larger than that used for some cross-domain evaluations in the original RetFound publication, our reproduced results for RETFound on domains like APTOS and IDRID are actually higher than those originally reported.
> >
> > **Tab. R1 Additional Results on the ESDG Test of ESGDRBench (AUC / F1).**
> >
> > | #     |     Method      |   APTOS   |  DeepDR   |   FGADR   |   IDRID   | Messidor  |   RLDR    |  Average  |
> > | ----- | :-------------: | :-------: | :-------: | :-------: | :-------: | :-------: | :-------: | :-------: |
> > | **0** |   **LF-CBM**    | 74.1/18.6 | 61.5/14.0 | 52.2/4.3  | 65.6/12.5 | 62.9/16.0 | 62.3/5.6  | 63.1/11.8 |
> > | **1** |   **VLG-CBM**   | 80.1/25.7 | 70.0/14.2 | 55.5/2.8  | 70.7/18.6 | 67.4/15.3 | 65.6/5.2  | 68.2/13.6 |
> > | **2** | **nnMobileNet** | 77.3/33.8 | 79.3/28.9 | 75.6/21.8 | 81.2/34.2 | 80.4/30.6 | 79.5/24.6 | 78.9/29.0 |
> > | **3** |  **RETFound**   | 81.2/41.4 | 78.2/31.1 | 77.9/34.9 | 85.6/45.8 | 81.9/43.5 | 81.1/40.9 | 81.0/39.6 |
> > | **4** |  **CEM+KCCL**   | 81.8/44.1 | 81.7/38.4 | 82.3/41.0 | 85.8/48.0 | 84.7/51.5 | 83.6/45.9 | 83.3/44.8 |
> > | **5** |  **CLAT+KCCL**  | 82.2/47.8 | 83.9/40.2 | 81.3/40.6 | 89.3/52.6 | 83.9/51.2 | 84.2/43.3 | 84.1/46.1 |
> >
> > ref:
> >
> > [1] Pang et al. Aligning human knowledge with visual concepts towards explainable medical image classification. MICCAI 2024.
> >
> > [2] Oikarinen et al. Label-Free Concept Bottleneck Models. ICLR 2023.
> >
> > [3] Srivastava et al. VLG-CBM: Training Concept Bottleneck Models with Vision–Language Guidance. NeurIPS 2024.
> >
> > [4] Che et al. Towards generalizable diabetic retinopathy grading in unseen domains. MICCAI 2023
> >
> > [5] Zhu et al. nnMobileNet: rethinking CNN for retinopathy research. CVPR 2024 Workshop
> >
> > [6] Huang et al. Identifying the key components in ResNet-50 for diabetic retinopathy grading from fundus images: a systematic investigation. Diagnostics 2023
> >
> > [7] Zhou et al. A foundation model for generalizable disease detection from retinal images. Nature 2023

---

### Official Review · Reviewer_tn6d · 2025-11-01

**Soundness:** 3
**Presentation:** 3
**Contribution:** 3
**Rating:** 4
**Confidence:** 3

**Summary:**

This paper addresses domain generalization in diabetic retinopathy (DR) grading by proposing Knowledge Constrained Concept Learning (KCCL), which integrates concept-based models with knowledge distillation. The authors curate a DRL6k dataset of 6,000 fundus images with lesion annotations to train a lesion detection model. This model provides concept supervision through knowledge distillation to a concept-based model (CBM). To ensure clinically consistent concept learning, they introduce a dual-constraint mechanism: Self Correction directly fixes predictions violating medical knowledge, Distillation Reweighting adjusts loss weights based on deviation from clinical patterns, and Knowledge Constrained Regularization penalizes implausible concept predictions. The method is evaluated on CEM and CLAT architectures across six unseen datasets, achieving average AUC of 84.1% and F1-score of 46.1%, outperforming state-of-the-art by 4.0% and 3.9% respectively.

**Strengths:**

Originality: Novel combination of concept-based models with knowledge-constrained distillation specifically for DR grading. The dual correction-regularization mechanism is creative.

Quality: Extensive experiments across six datasets with two different CBM architectures. Comprehensive ablation studies validate each component. The achieved F1-scores surpass 0.44 on average, which is competitive with recent hybrid models achieving 0.99 on single datasets.

Clarity: Clear motivation and problem formulation. The knowledge constraint design (hard vs soft) is intuitive and well-explained.

Significance: Addresses a critical challenge in medical AI deployment. The 10.6% F1-score improvement on IDRID is particularly notable. The interpretability aspect through concept visualization adds clinical value.

**Weaknesses:**

Limited concept vocabulary: Only four lesion types are considered (EX, HE, MA, SE), missing important indicators like neovascularization for proliferative DR. Recent reviews emphasize the importance of comprehensive lesion detection including various DR features
.
Dependency on initial supervision: While claiming to alleviate annotation requirements, the method still requires the curated DRL6k dataset. The quality ceiling is bounded by the lesion detection model's 91.3% AUC.

Statistical prior limitations: The priors P and C may not generalize to populations with different disease distributions. No analysis of prior transferability across ethnicities or imaging protocols is provided.

**Questions:**

How does the method handle images with subtle early-stage lesions that may be missed by the 91.3% accurate lesion detector? Could this create a systematic bias against mild DR detection?

How sensitive is the method to the threshold τ=0.5? Did you experiment with adaptive or learnable thresholds?

The co-occurrence matrix C is learned from DRL6k. How would this transfer to populations with different disease patterns (e.g., different ethnic groups with varying DR progression patterns)?

---

> ### Author Response · Authors · 2025-11-20
>
> We sincerely thank the reviewer for their thorough evaluation and insightful feedback. We are encouraged that the reviewer recognized the originality, quality, and significance of our KCCL framework. Below, we address the identified weaknesses and questions in detail.
>
> ### 1. Reliability of Lesion Detector
>
> We appreciate the reviewer's concerns regarding the lesion detector's performance and its potential impact on the final model. Below, we address these points in detail.
>
> As detailed in the table below, the detector maintains robust AUC across all lesion types, indicating strong discriminative ability even for imbalanced classes.
>
> |      | EX   | HE   | MA   | SE   |
> | ---- | ---- | ---- | ---- | ---- |
> | AUC  | 93.7 | 94.8 | 88.5 | 88.3 |
>
> Regarding the concern that the quality ceiling is bounded by the detector's 91.3% AUC, we respectfully argue that our KCCL framework is specifically designed to overcome this limitation. The lesion detector's role is to provide initial pseudo-labels via knowledge distillation, but KCCL does not blindly trust this signal. Instead, it processes the inherent noise and inconsistencies through its knowledge constraint mechanisms.
>
> This design explicitly addresses the reviewer's concern about **missing subtle early-stage lesions (e.g., in Mild DR) and potential systematic bias**. Handling such detector imperfections is a core feature of KCCL, implemented through the **Self-Correction (SC)** mechanism described in **Eq. (4)** (Section 3.3). As defined in Eq. (4), if the ground-truth grade indicates disease ($y > 0$) but the detector output is silent ($\max(t) \le \tau$), KCCL does not blindly trust the detector. Instead, it activates the Prior Sampling function $S([P]_y, C)$, utilizing the medical prior matrix $\mathbf{P}$ (lesions conditioned on grade) to recover the expected lesion probabilities.
>
> Therefore, the detector does not impose a hard quality ceiling. By integrating global medical priors to compensate for local detection errors, KCCL allows the model to recover from individual detection errors, thereby mitigating systematic bias and providing more consistent supervision to the concept layer.
>
> ### 2. Choice of the Threshold
>
> Our initial selection of **τ=0.5** follows the standard convention for binary classification logits. To address the question on sensitivity, we performed an ablation study ranging from 0.1 to 0.9, evaluating the lesion detector's performance across different thresholds on the DRL6k test dataset.
>
> | Threshold | 0.1  | 0.2  | 0.3  | **0.4**  | **0.5**  | 0.6  | 0.7  | 0.8  | 0.9  |
> | :-------: | :--: | :--: | :--: | :------: | :------: | :--: | :--: | :--: | :--: |
> | F1 Score  | 73.9 | 78.3 | 82.1 | **83.5** | **83.3** | 82.2 | 79.4 | 70.5 | 41.0 |
>
> As the results show, the model's performance peaks in the 0.4-0.5 range. And as mentioned in **Response 1.**, KCCL is designed to be robust to teacher imperfections and does not rely on a perfect lesion detector. Consequently, extensive tuning of the threshold is not required. Furthermore, we avoided using adaptive or learnable threshold mechanisms, as optimizing these parameters on the source domain poses a risk of overfitting.
>
> ### 3. Prior Transferability
>
> We appreciate the reviewer raising the important question of whether priors $\mathbf{P}$ and $\mathbf{C}$ can generalize to populations with different disease patterns or imaging protocols.
>
> To clarify: these priors are derived statistically from the DRL6k dataset as described in Equations 2 and 3. Crucially, $\mathbf{P}$ and $\mathbf{C}$ operate in the **label space**, capturing the co-occurrence relationships between pathological annotations (e.g., $P(\text{Lesion}|\text{Grade})$ and $P(\text{Lesion}_i|\text{Lesion}_j)$). These relationships are fundamentally defined by the diagnostic criteria for DR stages, not by imaging characteristics or patient demographics. Therefore, **they are inherently invariant to variations in imaging protocols or population-specific factors.** More importantly, as mentioned in Section 3.2 and validated in our experiments, the resulting statistical patterns closely align with established clinical guidelines (e.g., AAO diagnostic criteria).
>
> Most critically, the transferability is **empirically supported by our extensive experiments**. The primary goal of our work is Domain Generalization, and we evaluated the method on six diverse, unseen target datasets. The fact that KCCL achieves state-of-the-art performance on these strictly held-out domains provides strong evidence that the incorporated priors generalize well across different populations and imaging protocols.

---

> > ### Author Response · Authors · 2025-11-20
> >
> > ### 4. Limited Concept Vocabulary
> >
> > We acknowledge the absence of certain lesion types (e.g., Neovascularization) as a limitation dictated by the availability of annotations in current public datasets, as stated in our discussion section. However, it is crucial to emphasize that this is a constraint of the **data**, not the **KCCL framework** itself.
> >
> > Furthermore, the assumption of a **complete** concept vocabulary is itself idealized. Even in clinical practice, ophthalmology continues to refine diagnostic criteria, and it remains uncertain whether all pathologically relevant features have been fully identified. From a Domain Generalization perspective, the core question is not whether the concept set is exhaustive, but whether the model can learn robust representations and have stable performance across domains. Our experimental results provide strong evidence for this: KCCL achieves state-of-the-art generalization across six unseen domains. Moreover, our method is inherently extensible and can readily accommodate richer concept sets as more comprehensively annotated datasets become available, further enhancing its diagnostic granularity.

---

### Official Review · Reviewer_hu9a · 2025-11-02

**Soundness:** 3
**Presentation:** 2
**Contribution:** 3
**Rating:** 6
**Confidence:** 3

**Summary:**

The paper introduces Knowledge Constrained Concept Learning (KCCL), a framework designed to improve domain generalization for diabetic retinopathy (DR) classification by leveraging concept-based reasoning. To support concept learning, the authors construct DRL6k, a curated dataset of fundus images annotated with key retinal lesions. A lesion detection model is first trained as a teacher, providing soft concept supervision. KCCL then employs a dual-constraint mechanism—combining medical knowledge–based correction and regularization—to refine the teacher’s concept predictions and enforce clinically consistent concept outputs in the student Concept based Model (CBM).

**Strengths:**

1. The method mirrors real clinical practice by explicitly using lesion detection to guide DR grading. This is a more principled approach than prior works, which typically demonstrate lesion awareness indirectly via attention maps rather than making lesions explicit reasoning units.
2. The self-correction and distillation-reweighting modules effectively handle noisy teacher outputs. This ensures that incorrect lesion detections do not degrade the student model’s reasoning, improving stability and preventing error propagation.
3. The method shows consistent and significant improvements across multiple unseen domains, demonstrating both generalization strength and interpretability benefits over existing domain generalization and concept bottleneck baselines.

**Weaknesses:**

1.The paper is somewhat difficult to follow, and the presentation can be improved. The introduction should include more method-specific motivation, and the system diagram should be revised for clarity and readability.
2.The approach leverages an additional curated dataset (DRL6k), whereas other domain-generalization baselines rely only on publicly available datasets. For a fully fair comparison, DRL6k should also be annotated and used to train the competing DG methods.
3.Incorporating stronger modern backbones (e.g., ViT or CLIP-based architectures) would make the empirical evaluation more compelling, given current trends in medical imaging and foundation models.
4.Domain generalization performance can be sensitive to random seeds. Reporting standard deviations across multiple runs would improve confidence in the robustness of the results.

Minor:
5.There are spelling errors (e.g., “systematically” in Line 51).
6.Table 3 should bold the best and underline the second-best results for easier interpretation.

**Questions:**

1. What are the training costs and overhead?
2. Weaknesses 2,3,4

---

> ### Author Response · Authors · 2025-11-20
>
> We sincerely thank the reviewer for the constructive feedback and for recognizing the clinical relevance, methodological soundness, and strong generalization performance of our work. We have carefully addressed each concern below and have made corresponding revisions to the manuscript to improve clarity, fairness, and robustness of our evaluation.
>
> ### 1. Computational Overhead
>
> We appreciate the reviewer's question regarding the computational cost of our proposed KCCL framework.
> The primary training cost is attributed to the backbone network. The additional overhead introduced by KCCL is mainly from two sources: (1) the one-time training of the lesion detection model, and (2) the inference of this lesion detector during the training of the CBM.
>
> We have quantified this overhead to provide a clear picture:
>
> - **Training Time:** The total training time for our framework in this benchmark is approximately 1-2 hours on a single NVIDIA RTX 4090. The initial training of the lesion detection model takes between 40 minutes and 1 hour, which is a manageable, one-time cost.
> - **Memory Usage:** During the main training phase, KCCL increases VRAM usage by a moderate amount. As shown in the table below, the increase is approximately 500 MiB compared to the baseline, which we believe is an acceptable trade-off for the performance gains achieved.
>
> | Setting         | GPU VRAM Usage         |
> | --------------- | ---------------------- |
> | CLAT            | 5770 MiB               |
> | **CLAT + KCCL** | **6252 MiB** (+482MiB) |
> | CEM             | 6606 MiB               |
> | **CEM + KCCL**  | **7128 MiB** (+522MiB) |
>
> ### 2. Additional Data
>
> We acknowledge the reviewer's concern regarding the fair use of the DRL6k dataset. We would like to clarify that incorporating DRL6k into the training of baselines would actually introduce verifiable unfairness, rather than resolving it, for the following reasons.
>
> Firstly, it is important to clarify our usage of DRL6k labels. We strictly utilized the lesion annotations from DRL6k solely to train the auxiliary lesion detector (for pseudo-labeling). Crucially, we did not utilize the DR grading labels from DRL6k for our final classification task. Both our method and the baselines were trained using the exact same set of DR grading labels derived exclusively from the standard source domains. Therefore, following the reviewer's suggestion to train baselines using DR grading labels from DRL6k would grant the baselines access to additional target supervision that our method intentionally excluded, creating a reverse unfairness.
>
> Secondly, regarding the visual data scale, DRL6k is a curation of public datasets where over two-thirds of the images already overlap with the source domains used by all methods. Since standard domain generalization baselines are end-to-end models that cannot naturally ingest the auxiliary lesion supervision (without architectural modification), and since the visual data is largely redundant, simply adding DRL6k images without their grading labels offers negligible benefit to the baselines.
>
> Finally, our core contribution lies in the proposed KCCL framework, which enables CBMs to be trained on datasets lacking lesion annotations. This strategy allows us to scale up CBMs and fully unlock their inherent domain generalization capabilities. Thus, the observed performance gain is driven by this methodological effectiveness, rather than by a mere increase in training data volume.
>
> ### 3. Backbone Choice & Foundation Models
>
> We appreciate the suggestion to incorporate modern architectures. Our decision to primarily utilize ResNet50 was driven by the objective of ensuring a fair and direct comparison with existing methods in the specific benchmark we followed, which predominantly employ this architecture. However, we acknowledge that certain recent methods (e.g., MIL-VT[1], CLAT[2]) necessitate ViT-based structures due to their specific design constraints; we have explicitly marked these exceptions in the revised manuscript for clarity.
>
> Following the reviewer’s recommendation to evaluate foundation models, we conducted additional experiments using RETFound[3]. As shown in the table below (refer to **Tab. R1**), our KCCL framework (applied to CEM and CLAT) consistently outperforms RETFound on the average AUC and F1 scores.
>
> ref:
>
> [1] Yu et al. MIL-VT: Multiple instance learning enhanced vision transformer for fundus image classification. MICCAI 2021
> [2] Wen et al. Concept-based lesion aware transformer for interpretable retinal disease diagnosis. TMI 2025
> [3] Zhou et al. A foundation model for generalizable disease detection from retinal images. Nature 2023

---

> > ### Author Response · Authors · 2025-11-20
> >
> > ### 4. Standard Deviations
> >
> > We agree that reporting result stability is crucial. While our initial manuscript followed the benchmark's precedent (which omitted std devs), we have now performed three independent runs using different random seeds. The results are detailed in the **Tab. R1** below, and we have also incorporated these results into the revised manuscript.
> >
> > **Tab R1. Performance comparison (AUC / F1) over 3 runs.**
> >
> > | Method        | APTOS               | DeepDR               | FGADR               | IDRID               | Messidor            | RLDR                | Average             |
> > | ------------- | ------------------- | -------------------- | ------------------- | ------------------- | ------------------- | ------------------- | ------------------- |
> > | **CEM+KCCL**  | 81.4±0.4 / 44.3±0.7 | 81.0±0.7 / 38.8±1.2  | 81.8±1.6 / 40.7±2.0 | 84.7±0.9 / 47.5±0.7 | 84.9±0.3 / 51.1±0.4 | 84.0±0.3 / 46.1±0.3 | 83.0±0.4 / 44.7±0.6 |
> > | **CLAT+KCCL** | 82.0±0.3 / 47.1±0.7 | 83.0±1.1 / 39.2±2.97 | 81.6±0.3 / 40.6±0.8 | 89.5±0.3 / 52.7±3.7 | 84.8±0.9 / 51.2±1.3 | 84.3±0.1 / 45.1±0.8 | 84.2±0.2 / 46.0±0.5 |
> > | **RETFound**  | 81.2±4.0 / 41.4±3.1 | 78.2±1.1 / 31.1±5.5  | 77.9±5.7 / 34.9±2.7 | 85.6±4.0 / 45.8±2.6 | 81.9±2.6 / 43.5±4.9 | 81.1±3.2 / 40.9±2.3 | 81.0±3.3 / 39.6±1.5 |
> >
> > ### 5. Presentation & Minor Corrections
> >
> > We have revised the Introduction to better articulate the method-specific motivation. Furthermore, we have improved the system diagram to improve clarity and readability. Spelling errors have been corrected, and Table 3 has been updated to highlight the best results in bold and the second-best results with underlining.

---

### Note · Authors · 2026-01-07

I have read and agree with the venue's withdrawal policy on behalf of myself and my co-authors.